# Comparison of the GLIM, ESPEN and ICD-10 Criteria to Diagnose Malnutrition and Predict 30-Day Outcomes: An Observational Study in an Oncology Population

**DOI:** 10.3390/nu13082602

**Published:** 2021-07-28

**Authors:** Shay Poulter, Belinda Steer, Brenton Baguley, Lara Edbrooke, Nicole Kiss

**Affiliations:** 1School of Exercise and Nutrition Sciences, Deakin University, Melbourne 3125, Australia; spoulter@deakin.edu.au; 2Nutrition and Speech Pathology Department, Peter MacCallum Cancer Centre, Melbourne 3000, Australia; Belinda.Steer@petermac.org; 3Institute for Physical Activity and Nutrition, Deakin University, Geelong 3220, Australia; b.baguley@deakin.edu.au; 4Allied Health Department, Peter MacCallum Cancer Centre, Melbourne 3000, Australia; Lara.Edbrooke@petermac.org; 5Physiotherapy Department, The University of Melbourne, Parkville 3052, Australia

**Keywords:** GLIM, ESPEN, ICD-10, malnutrition, cancer, predictive validity, mortality, unplanned admission

## Abstract

The Global Leadership Initiative on Malnutrition (GLIM) criteria are consensus criteria for the diagnosis of malnutrition. This study aimed to investigate and compare the prevalence of malnutrition using the GLIM, European Society for Clinical Nutrition and Metabolism (ESPEN) and International Statistical Classification of Diseases version 10 (ICD-10) criteria; compare the level of agreement between these criteria; and identify the predictive validity of each set of criteria with respect to 30-day outcomes in a large cancer cohort. GLIM, ESPEN and ICD-10 were applied to determine the prevalence of malnutrition in 2794 participants from two cancer malnutrition point prevalence studies. Agreement between the criteria was analysed using the Cohen’s Kappa statistic. Binary logistic regression models were used to determine the ability of each set of criteria to predict 30-day mortality and unplanned admission or readmission. GLIM, ESPEN and ICD-10 criteria identified 23.0%, 5.5% and 12.6% of the cohort as malnourished, respectively. Slight-to-fair agreement was reported between the criteria. All three criteria were predictive of mortality, but only the GLIM and ICD-10 criteria were predictive of unplanned admission or readmission at 30 days. The GLIM criteria identified the highest prevalence of malnutrition and had the greatest predictive ability for mortality and unplanned admission or readmission in an oncology population.

## 1. Introduction

People with cancer are 1.7 times more likely to be malnourished than other hospitalised patients [1]. Cancer-related malnutrition affects between 30% and 70% of people with cancer, with the prevalence varying by cancer type, treatment modality and method of assessing malnutrition [2,3,4]. The adverse outcomes associated with cancer-related malnutrition have been widely reported, including, but not limited to, a decrease in treatment tolerance and increased mortality, hospital admission/readmission and length of hospital stay, leading to greater healthcare costs [3,5,6,7,8,9,10,11,12]. Early identification, timely assessment and treatment of malnutrition are recommended to facilitate optimal care and minimise the adverse outcomes associated with cancer-related malnutrition [13].

Three criteria used internationally to diagnose malnutrition include the Global Leadership Initiative on Malnutrition (GLIM) criteria [14], European Society for Clinical Nutrition and Metabolism (ESPEN) criteria [15] and International Statistical Classification of Diseases and Related Health Problems, Tenth Revision (ICD-10) [16]. There are differences in each of these criteria that influence the diagnosis of malnutrition. The GLIM criteria, developed in 2018, consider both phenotypic criteria including body mass index (BMI) with cut-offs by age, unintentional weight loss in a specified timeframe and low muscle mass, and aetiological criteria including reduced food intake/assimilation over a specified timeframe and inflammation [14]. In 2015, ESPEN developed criteria for malnutrition diagnosis, which include weight loss in a specified timeframe; BMI with cut-offs by age; and fat-free mass index (FFMI), with two diagnostic options, with and without fat-free mass. However, unlike ICD-10, ESPEN does not consider food intake in the criteria [15]. ICD-10 includes the assessment of BMI, weight loss, food intake and muscle stores; however, it lacks detail regarding a timeframe for weight loss or the degree of reduced food intake [16]. The GLIM criteria are proposed by the GLIM Core Leadership Committee and the GLIM Working Group as the global diagnostic criteria, to establish and improve standardisation of malnutrition diagnosis and comparison across studies and clinical populations. However, the GLIM criteria require validation in various clinical populations to support use and global uptake of the criteria [17].

Recently, the GLIM criteria have been suggested as a replacement for the current ICD-10 malnutrition criteria in the 11th revision of ICD [14]. Valid and robust ICD criteria for malnutrition diagnosis are critical since ICD determines hospital funding and reimbursement related to the treatment of malnutrition [18]. Studies comparing the validity of the current ICD-10 malnutrition criteria with other commonly used criteria are important to support advocacy for an update of the ICD malnutrition criteria. While the ability of the GLIM criteria to predict adverse outcomes in oncology populations using objective measures of muscle mass, they have been the focus of some studies [19,20], a comparison of the predictive validity of these criteria to ICD-10 has not previously been published. Furthermore, the ability to perform objective measurement of muscle mass in clinical practice for the GLIM and ESPEN criteria is dependent upon access to the appropriate equipment and training. Therefore, comparing the predictive validity of these criteria, using the more feasible subjective physical assessment of muscle stores, will support establishing validity in the form that these criteria are likely to be used in clinical practice for the assessment of malnutrition. The aims of this study in an adult oncology cohort were (1) to determine malnutrition prevalence according to the GLIM, ESPEN and ICD-10 criteria, (2) compare the level of agreement in malnutrition diagnosis between each criteria and (3) to determine the predictive validity of a diagnosis of malnutrition according to each criteria with respect to mortality and unplanned admission or readmission at 30 days post data collection.

## 2. Materials and Methods

### 2.1. Study Design and Setting

This study is a secondary data analysis using the collected data from two multicentre observational point prevalence studies (PPS) conducted in 2016 and 2018, with the methodology previously reported [21].

A total of 16 and 19 public and private health services within Victoria, Australia, participated in the study in 2016 and 2018, respectively. Recruitment occurred over a four-week period during November–December of 2016 and July–August of 2018. Sites collected data from multi-stay inpatient, chemotherapy day unit and outpatient radiotherapy services. Site dietitians or student dietitians who had reached clinical competency completed data collection, including malnutrition screening.

### 2.2. Participants and Eligibility

Eligible patients were aged ≥ 18 years; with a diagnosis of cancer, admitted to hospital (minimum two nights) for cancer treatment and/or related management, or outpatients attending for radiotherapy or intravenous chemotherapy or immunotherapy. Patients were excluded if they were admitted to an Intensive Care Unit (ICU) or emergency department on the day of data collection; were attending for day surgery, medical review, oral chemotherapy, or maintenance/hormonal therapy only; were terminally ill with a life expectancy less than one month; or were unable to provide verbal consent. All patients provided verbal consent to participate prior to data collection. Multi-site ethics approval was received through Peter MacCallum Cancer Centre Ethics Committee (HREC/16/PMCC149) for the original PPS. Exemption from ethics approval for this secondary analysis was approved by Deakin University (2020-172).

### 2.3. Demographic and Clinical Data

Demographic variables included: age, sex and current living situation (alone, with family or in residential care). Clinical data included the primary malignancy, the presence of metastasis, the treatment setting (inpatient, day patient or outpatient) and current treatment (including surgery, chemotherapy, radiotherapy, stem cell transplant, immunotherapy or cancer-related management). At 30 days following the original data collection, unplanned admission, or readmission, of more than two nights to the same hospital and patient status (alive, deceased or unknown) were collected from medical records.

### 2.4. Malnutrition Screening

Participants were screened using the Malnutrition Screening Tool (MST), which is validated for use in oncology populations [22,23,24]. Participants with an MST score of ≥2 were deemed at risk of malnutrition, while those with a score of <2 were deemed to be well nourished.

### 2.5. Anthropometry

Height (cm) and weight (kg) were measured using standard equipment at each hospital site or reported by participants and used to calculate BMI. Participants were asked whether they had lost weight, either intentionally or unintentionally, the degree of weight loss (kg) and over what timeframe it happened (≤3 months or ≥4 months).

### 2.6. Food Intake

Participants were asked whether they had experienced reduced food intake. If reduced food intake was reported, the extent of the reduction compared to usual food intake (>75%, ≤75%, ≤50%, ≤25%) and the duration (0–4 days, 5–30 days or ≥1 month) were recorded.

### 2.7. Nutrition Assessment

Participants at risk of malnutrition (MST score ≥ 2) underwent a subjective physical examination of muscle stores on a minimum of four out of seven muscle sites (temple, clavicle, shoulder, interosseous muscle, scapula, thigh, calf) from the valid and reliable Patient-Generated Subjective Global Assessment (PG-SGA) [25].

### 2.8. Malnutrition Diagnosis

Appendix A describes how the GLIM, ESPEN and ICD-10 criteria were applied using data from the present study. An objective measure of inflammation was not available for the GLIM criteria. Operational guidelines for validation of the GLIM criteria state it is not sufficient to assume inflammation is associated with all diagnoses of malignant disease and, therefore, the presence of metastatic disease was used as a proxy measure of inflammation/disease burden [26]. Data were only available to determine malnutrition diagnosis according to the ESPEN criteria without FFMI.

### 2.9. Statistical Analysis

Statistical analysis was conducted using SPSS Statistics version 26.0 (IBM, Armonk, NY, USA). Descriptive statistics were used to summarise demographic and clinical characteristics. Continuous variables were presented as mean and standard deviations and categorical variables as number and percentages. Agreement between the cancer-related malnutrition criteria using Cohen’s kappa coefficient was classified as slight (≤0.20), fair (0.21–0.40), moderate (0.41 to 0.60), substantial (0.61–0.80) and excellent (>0.80) [27].

Binary logistic regression models were used to assess the predictive validity of each malnutrition diagnostic criteria for 30-day outcomes of mortality and unplanned admission or readmission. Assumptions of collinearity were assessed. The covariates considered for inclusion in the models were age (continuous), sex, living situation, patient type, treatment hospital location and presence of metastasis. Separate models were performed for each of the malnutrition diagnostic criteria, and the models with the highest predictive validity and including only variables making a statistically significant (*p* < 0.05) contribution to the model were reported.

## 3. Results

A total of 2801 participants met the inclusion criteria for the original point prevalence study from 2016 and 2018 in the combined datasets. Of the 2801 participants, 7 had insufficient data to be classified as at-risk or not at risk by MST and were excluded, leaving 2794 who were included in the analysis (Figure 1). Demographic and clinical characteristics are described in Table 1. The mean age was 62.7 years (±14.1), with just over half considered elderly (≥65 years), and a mean BMI of 27.0 kg/m^2^ (±5.9). The majority of participants were outpatients (77.1%), living with family/carer/in residential care (81.2%) and receiving chemotherapy (68.7%). The highest proportion of cancer types were haematological (18.8%), breast (17.9%) and colorectal cancer (14.0%), and 1003 participants (35.9%) presented with metastatic disease.

### 3.1. Malnutrition Prevalence by GLIM, ESPEN and ICD-10

Figure 1 shows that overall, 35.1% (981/2794) of participants were classified as ‘at risk’ by the MST. The prevalence of malnutrition was 23.0% (616/2679) by GLIM, 5.5% (*n* = 149/2691) by ESPEN and 12.6% (351/2778) by ICD-10. Of the 616 participants diagnosed as malnourished by GLIM, 320 (51.9%) were classified as severely malnourished. Missing data precluded the application of the criteria for some participants as shown in Figure 1. Table 2 describes the prevalence of the individual assessment criteria for the GLIM, ESPEN and ICD-10 criteria.

### 3.2. Agreement between Malnutrition Criteria

The agreement between the malnutrition criteria is shown in Table 3. The agreement between the GLIM and ESPEN criteria was slight (Kappa 0.07, 95% CI 0.04–0.10, *p* < 0.001). There was also fair agreement between the GLIM and ICD-10 criteria (Kappa 0.34, 95% CI 0.28–0.39, *p* < 0.001). While the agreement between the ESPEN and ICD-10 criteria was fair (Kappa 0.33, 95% CI 0.03–0.39, *p* < 0.001).

### 3.3. Prediction of 30-Day Mortality

Results of the logistic regression models are presented in Table 4. A total of 87 (3%) participants were deceased at 30 days post data collection, and 63 (72%) of these participants were identified as being at risk of malnutrition. Of the participants who were deceased at 30 days, 48% (42/87) were malnourished according to GLIM, 16 (14/87) were classified as malnourished by ESPEN and 30% (26/87) were classified as malnourished by ICD-10. All three criteria were predictive of mortality at 30 days (OR (95% CI)); GLIM 2.53 (1.46–4.39), *p* < 0.001; ESPEN 2.01 (1.02–3.98), *p* < 0.001; and ICD-10 2.34 (1.34–4.10), *p <* 0.001. Further detail is provided in Appendix A.

### 3.4. Prediction of 30-Day Unplanned Admission or Readmission

A total of 354 (13%) participants experienced an unplanned admission or readmission within 30 days of data collection, and 184 (52%) of those were at risk of malnutrition. Of the total 354 participants who experienced unplanned hospital admission or readmission, the GLIM criteria classified 36% (126/354) as malnourished, ESPEN classified 8% (29/354) as malnourished and ICD-10 classified 22% (77/354) as malnourished. Only the GLIM and ICD-10 criteria were predictive of unplanned admission or readmission within 30 days (OR (95% CI)); GLIM 1.76 (1.33–2.32), *p* < 0.001; and ICD-10 1.57 (1.13–2.20), *p* < 0.001. A diagnosis of malnutrition using the ESPEN criteria was not associated with a greater likelihood of unplanned admission or readmission within 30 days (OR (95% CI)) 1.08 (0.65–1.79), *p* = 0.77. Only a small variance could be explained by each model (Table 4), indicating factors that were not measured in this study contributed to unplanned admission or readmission.

The full regression models were significant, indicating the models were able to predict mortality and unplanned admission or readmission. All models had high specificity (100%) as all participants who were alive or did not experience unplanned admission or readmission were correctly identified, but low sensitivity (0%) as all models predicted zero patients deceased or with unplanned admission or readmission at 30 days.

## 4. Discussion

In this study, involving a large oncology cohort, the GLIM criteria identified a higher proportion of patients as malnourished in comparison to the ICD-10 and ESPEN criteria. Participants who were malnourished by any diagnostic criteria were more likely to be deceased at 30 days. However, only participants who were malnourished in accordance with the GLIM and ICD-10 criteria were more likely to have an unplanned admission or readmission within 30 days of data collection. To our knowledge, this is the first study to explore the agreement between the GLIM, ESPEN and ICD-10 criteria for the diagnosis of malnutrition using pragmatic muscle assessment within any clinical population. It is also the first study to compare the predictive ability of these criteria for 30-day mortality and unplanned admission or readmission within an oncology setting.

There was substantial variation in malnutrition prevalence according to the three diagnostic criteria ranging from 23% for GLIM and 5.5% for ESPEN to 12.6% for ICD-10. The prevalence of malnutrition from this study was lower compared to that in previous studies over the past ten years, ranging from 26% to 73% when using various diagnostic tools in cancer populations [3,4,5,28,29,30,31,32]. However, the prevalence of malnutrition diagnosed using the GLIM criteria in our study was comparable to the prevalence of 26–31% in a similar cancer population where malnutrition was assessed using the Patient-Generated Subjective Global Assessment (PG-SGA) [3]. Our results are consistent with a study conducted in a geriatric non-cancer population where GLIM also identified a higher proportion of participants as malnourished compared to ESPEN (52% vs. 12.6%), despite using the ESPEN criteria with FFMI [33].The substantially lower prevalence using the ESPEN criteria may be explained by our study not including FFMI, where participants were required to have a BMI of less than 20 kg/m^2^ or 22 kg/m^2^ if under 70 or 70 years or older, respectively [15]. This resulted in participants with substantial weight loss and muscle deficits being classified as well nourished if their BMI was over this threshold. For a classification of malnutrition using ICD-10, a BMI of less than 18.5 kg/m^2^ or all three parameters of weight loss, reduced food intake and muscle deficit needed to be present [16]. In participants with a BMI of more than 18.5 kg/m^2^, this led to participants who met the criteria for either, but not both, weight loss and muscle deficit plus reduced food, being classified as well nourished. This limits the ability to identify malnourished patients who are overweight or obese yet have significant loss of weight and/or muscle deficits. The flexibility of the GLIM criteria to diagnose malnutrition through the presence of one of three phenotypic and one of two aetiological criteria led to a higher proportion of the sample being classified as malnourished [14].

Agreement between the criteria was poor to fair and particularly low between the ESPEN and GLIM criteria. In addition to the above-mentioned differences in diagnostic requirements, there are several further reasons for this relatively low agreement. The requirement of at least one phenotypic criterion and one aetiologic criterion for a malnutrition diagnosis using the GLIM criteria meant participants, who in some cases met the severe grading for a phenotypic criterion but not one of the aetiologic criterion, were classified as well nourished [14]. This may be due to the use of metastatic disease as a proxy measure of inflammation in this study, and it is possible these participants may have been classified as malnourished had an objective measure of inflammation been available. Using the ESPEN and ICD-10 criteria, a diagnosis of malnutrition can be made when BMI is less than 18.5 kg/m^2^ alone in the absence of aetiologic criteria. Furthermore, although both the ESPEN and ICD-10 criteria require the presence of 5% or more weight loss, the ESPEN criteria specify this must have occurred within three months, while ICD-10 does not specify a timeframe. The need for reduced food intake to also be present within ICD-10 could further account for poorer agreement with the ESPEN criteria but would increase agreement with the GLIM criteria. The GLIM criteria also require reduced food intake to have occurred for a certain duration, while ICD-10 does not require a timeframe. However, the advantages of the GLIM criteria are apparent in that the GLIM criteria were able to capture the vast majority of participants classified as malnourished by ICD-10 and ESPEN, while a relatively small proportion of participants diagnosed as malnourished by the GLIM criteria were captured by ESPEN and ICD-10.

Within this study, the odds of 30-day mortality from initial data collection ranged between 2.01 and 2.53 times higher in malnourished than in well-nourished patients, with malnutrition diagnosed by the GLIM, ESPEN and ICD-10 criteria all predictive of 30-day mortality in a cancer cohort. However, the association was greatest with a diagnosis of malnutrition using the GLIM criteria, which could be due to the range of phenotypic and aetiologic criteria underpinning the diagnosis. The GLIM criteria have previously been shown to be predictive of both six-month and one-year mortality in various cancer populations, in both inpatient and outpatient settings [10,19,20,34]. Similarly, Rondel et al. have shown that a diagnosis of malnutrition using the ESPEN criteria was also associated with increased risk of mortality in hospitalised patients, albeit using the version of the ESPEN criteria including FFMI [35]. It is possible that the ability of the ESPEN criteria to predict 30-day mortality in the current study may have been improved with the addition of FFMI. Malnutrition diagnosed by the GLIM and ICD-10 criteria, but not the ESPEN criteria, was associated with a greater likelihood of unplanned admission or readmission at 30 days. In contrast, a previous study in cancer inpatients by Contreras-Bolivar et al. did not find an association between hospital readmission and malnutrition diagnosed by the GLIM criteria using handgrip strength and anthropometric measures of muscle mass [19]. To our knowledge, no studies have previously investigated the association between malnutrition diagnosed using the ICD-10 criteria and mortality or unplanned admission or readmission.

The GLIM criteria recommend the measurement of muscle mass using a validated body composition technique such as dual-energy X-ray absorptiometry (DXA) or bio-electrical impedance analysis (BIA). However, these devices are not readily available within all clinical settings; therefore, subjective physical examination or anthropometric measures are suggested as suitable alternatives [14]. Previous studies have shown pragmatic muscle assessment using various anthropometrical measures within the GLIM diagnosis is predictive of mortality [20,36]. Similarly, our study has demonstrated that pragmatic muscle assessment using the physical assessment of the PG-SGA is also predictive of mortality and unplanned admission or readmission when used to diagnose malnutrition according to the GLIM and ICD-10 criteria, supporting its use in clinical practice.

This study has some limitations that should be noted. The secondary data analysis study design limited the inclusion of factors from previous studies known to increase the risk of cancer-related malnutrition including tumour stage and combined treatment modalities. As recommended by the GLIM criteria, a two-step process of risk screening followed by assessment was followed in this study. Participants classified as not at risk of malnutrition were considered well nourished, and no further assessment was undertaken to assess these patients by the three diagnostic criteria. It was noted that some participants who were screened as not at risk by the MST would have been identified as malnourished on the basis of a BMI <18.5 kg/m^2^. The classification of the GLIM aetiological criteria for inflammation through the presence of metastasis may have underestimated the true presence of inflammation in study participants. However, the GLIM consensus group recommend against considering the presence of malignant disease as representative of inflammation and recommend differentiating disease burden between participants [26]. Additionally, the low number of deceased patients within the total cohort of each model (2.9–3.4%) could have impacted the specificity (100%) and sensitivity (0%) within the regression models.

## 5. Conclusions

In this large cancer cohort, the GLIM criteria detected the highest prevalence of malnutrition and was associated with the greatest likelihood of mortality and unplanned admission or readmission at 30 days in comparison to the ESPEN and ICD-10 criteria. The findings from this study support the use of the GLIM criteria for the classification of malnutrition in the next update of ICD. However, as this is the first study comparing these diagnostic criteria, further studies are recommended in other clinical populations. Secondly, a comparison between the three criteria with the use of FFMI for the ESPEN criteria is also recommended to determine the effect on the agreement between malnutrition criteria and the predictive ability of each set of criteria.

## Figures and Tables

**Figure 1 nutrients-13-02602-f001:**
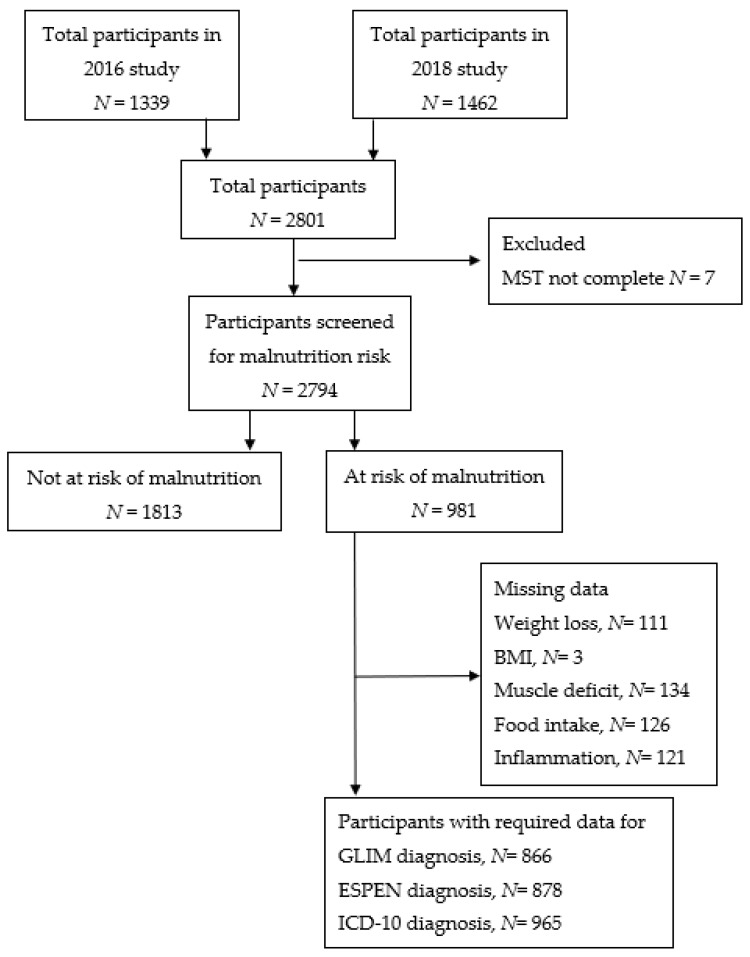
Participant flow diagram.

**Table 1 nutrients-13-02602-t001:** Demographic and clinical characteristics of participants (*N* = 2794).

	Total *N* = 2794 ^a^	Female *N* = 1396	Male *N* = 1397
	*N* (%)	*N* (%)	*N* (%)
Age (years)			
Mean (SD)	62.7 (14.1)	60.2(14.6)	65.2 (13.2)
18–49 years	470 (16.8)	319 (22.9)	151 (10.8)
50–64 years	917 (32.8)	483 (34.6)	434 (31.1)
65–79 years	1126 (40.3)	479 (34.3)	646 (46.2)
≥80 years	273 (9.8)	111 (8.0)	162 (11.6)
Unknown	8 (0.3)	4 (0.3)	4 (0.3)
Body mass index (kg/m^2^)			
Mean (SD)	27.0 (5.9)	27.2 (6.4)	26.9 (5.3)
<18.5	85 (3.0)	51 (3.7)	34 (2.4)
18.5–24.9	1038 (37.2)	531 (38.0)	507 (36.3)
25–29.9	957 (34.3)	425 (30.4)	531 (38.0
≥30	706 (25.3)	387 (27.7)	320 (22.9)
Unknown	8 (0.3)	2 (0.1)	5 (0.4)
Patient type			
Inpatient	638 (22.8)	284 (20.3)	354 (25.3)
Outpatient/day patient	2156 (77.2)	1112 (79.7)	1043 (74.7)
Hospital location			
Metropolitan	2377	1206 (86.4)	1171 (83.8)
Regional	417	190 (13.6)	226 (16.2)
Living situation			
Alone	517 (18.5)	275 (19.7)	242 (17.3)
Family/carer/residential	2270 (81.2)	1119 (80.2)	1150 (82.6)
Unknown	7 (0.3)	2 (0.1)	2 (0.1)
Primary malignancy			
Bone and soft tissue	51 (1.8)	18 (1.3)	33 (2.4)
Breast	501 (17.9)	497 (35.6)	4 (0.3)
Central nervous system	37 (1.3)	19 (1.4)	18 (1.3)
Colorectal	391 (14.0)	160 (11.5)	230 (16.5)
Endocrine and thyroid	47 (1.7)	26 (1.9)	21 (1.5)
Genitourinary	218 (7.8)	31 (2.2)	187 (13.4)
Gynaecological	144 (5.2)	141 (10.1)	3 (0.2)
Haematological	527 (18.9)	199 (14.3)	328 (23.5)
Head and neck	188 (6.7)	49 (3.5)	139 (9.9)
Lung	307 (11.0)	129 (9.2)	178 (12.7)
Skin and melanoma	132 (4.7)	41 (2.9)	91 (6.5)
Upper gastrointestinal	210 (7.5)	71 (5.1)	139 (9.9)
Other thoracic or abdominal	19 (0.7)	5 (0.4)	14 (1.0)
Cancer of unknown primary	18 (0.6)	8 (0.6)	10 (0.7)
Unknown	4 (0.1)	2 (0.1)	2 (0.1)
Presence of metastasis			
Yes	1003 (35.9)	494 (35.4)	509 (36.4)
No	1471 (52.6)	751 (53.8)	719 (51.5)
Uknown	320 (11.5)	151 (10.8)	169 (12.1)
Treatment type ^b^			
Chemotherapy	1920 (68.7)	1000 (71.6)	919 (65.8)
Radiotherapy	876 (31.4)	423 (30.3)	453 (32.4)
Surgery	655 (23.4)	377 (27.0)	277 (19.8)
Stem cell transplant	58 (2.1)	20 (1.4)	38 (2.7)
Immunotherapy	170 (6.1)	76 (5.4)	94 (6.7)
Other	211 (7.6)	88 (6.3)	123 (8.8)
Cancer-related management	76 (2.7)	37 (2.7)	39 (2.8)
MST ^c^ classification			
Not at risk of malnutrition (score < 2)	1813 (64.9)	955 (68.4)	858 (61.4)
At risk of malnutrition (score ≥ 2)	981 (35.1)	441 (31.6)	539 (38.6)

^a^ Sex was not reported for one participant; ^b^ some patients were receiving more than one treatment concurrently; ^c^ MST, malnutrition screening tool; SD, standard deviation.

**Table 2 nutrients-13-02602-t002:** Malnutrition prevalence and assessment criteria according to GLIM, ESPEN and ICD-10 diagnostic criteria (*N* = 2794).

Diagnostic Criteria	Cut-Off	Prevalence, *N* (%)
GLIM		
≥1 *Phenotypic criteria*	>5% weight loss in ≤3 months	619 (23.1)
Unintentional weight loss	>10% weight loss in ≥4 months	
Low BMI	<20 kg/m^2^ if <70 years	202 (7.5)
	<22 kg/m^2^ if ≥70 years	
Low muscle mass	Minimum of 4 muscle sites rated mild/moderate or severe	474 (17.7)
AND		
≥1 *Etiologic criteria*		
Reduced food intake	≤50% usual intake for 5–30 days or ≥1 month	678 (25.3)
	≤75% usual intake for ≥1 month	
	>75% usual intake for ≥1 month	
Inflammation	Presence of metastatic disease	399 (14.9)
GLIM malnutrition prevalence		616 (23.0)
GLIM moderate malnutrition prevalence		296 (11.0)
GLIM severe malnutrition prevalence		320 (11.9)
ESPEN		
*Option 1*		
Low BMI	<18.5 kg/m^2^	66 (2.4)
*Option 2*		
Unintentional weight loss	>5% weight loss in ≤3 months	619 (23.0)
AND	>10% weight loss in ≥4 months	
Low BMI	<20 kg/m^2^ if <70 years	202 (7.5)
	<22 kg/m^2^ if ≥70 years	
ESPEN malnutrition prevalence		149 (5.5)
ICD-10		
*Option 1*		
Low BMI	<18.5 kg/m^2^	66 (2.3)
*Option 2*		
Unintentional weight loss	>5% weight loss	619 (22.2)
AND		
Reduced food intake	Any reduction in food intake	812 (29.2)
AND		
Mild/moderate muscle wasting	Minimum of 4 muscle sites rated mild/moderate or severe	474 (17.0)
ICD-10 malnutrition prevalence		351 (12.6)

Abbreviations: GLIM, Global Leadership Initiative on Malnutrition criteria; ESPEN, European Society of Clinical Nutrition and Metabolism malnutrition criteria: ICD-10, International Classification of Disease version 10 malnutrition criteria, BMI Body Mass Index.

**Table 3 nutrients-13-02602-t003:** Agreement between ICD-10, ESPEN and GLIM malnutrition diagnostic criteria.

	**ESPEN CRITERIA**		
GLIM CRITERIA		Malnourished (143)	Not malnourished (710)
	Malnourished (604)	121 (20.0%) ^a^	483 (80.0%) ^a^
	Not malnourished (249)	22 (8.8%) ^b^	227 (91.2%) ^b^
	**ICD-10 CRITERIA**		
GLIM CRITERIA		Malnourished (344)	Not malnourished (465)
	Malnourished (573)	316 (55.1%) ^a^	257 (44.9%) ^a^
	Not malnourished (236)	28 (11.9%) ^b^	208 (88.1%) ^b^
	**ESPEN CRITERIA**		
ICD-10 CRITERIA		Malnourished (148)	Not malnourished (726)
	Malnourished (351)	123 (35%) ^c^	228 (65%) ^c^
	Not malnourished (523)	25 (4.8%) ^d^	498 (95.2%) ^d^

Abbreviations: GLIM, Global Leadership Initiative on Malnutrition criteria; ESPEN, European Society of Clinical Nutrition and Metabolism malnutrition criteria; ICD-10, International classification of disease version 10 malnutrition criteria. ^a^ Reported as a proportion of GLIM malnourished cases, ^b^ reported as a proportion of GLIM not malnourished cases.^c^ Reported as a proportion of ICD-10 malnourished cases, ^d^ reported as a proportion of ICD-10 not malnourished cases.

**Table 4 nutrients-13-02602-t004:** Multivariate logistic regression of the association between GLIM, ESPEN and ICD-10 malnutrition diagnosis and mortality/unplanned admission or readmission at 30 days.

	B	S.E	Wald	df	*p*	Odds Ratio	95% CI	Adjusted Odds Ratio ^b^	95% CI
Mortality at 30 days									
GLIM ^a^	0.93	0.28	10.91	1	0.001	2.53	1.46–4.39	2.50	1.44–4.35
Patient type	2.10	0.29	50.91	1	<0.001	8.16	4.58–14.52	8.10	4.55–14.43
Metastasis	1.37	0.31	20.15	1	<0.001	3.95	2.17–7.18	3.89	2.14–7.08
Constant	−5.60	0.33	291.12	1	<0.001	0.004		0.001	
ESPEN ^a^	0.70	0.35	4.04	1	0.044	2.01	1.02–3.98	2.03	1.03–4.01
Patient type	2.31	0.27	74.73	1	<0.001	10.04	5.95–16.93	9.97	5.91–16.83
Metastasis	1.66	0.28	35.30	1	<0.001	5.24	3.03–9.06	5.12	2.96–8.86
Constant	−5.45	0.31	311.52	1	<0.001	0.004		0.002	
ICD-10 ^a^	0.85	0.29	8.87	1	0.003	2.34	1.34–4.10	2.35	1.34–4.13
Patient type	2.05	0.29	49.64	1	<0.001	7.78	4.34–13.78	7.76	4.39–13.73
Metastasis	1.41	0.29	22.91	1	<0.001	4.08	2.29–7.26	4.03	2.26–7.16
Constant	−5.42	0.32	291.65	1	<0.001	0.004		0.002	
Unplanned admission or readmission at 30 days									
GLIM ^a^	0.56	0.14	15.98	1	<0.001	1.76	1.33–2.32	1.78	1.34–2.35
Patient type	0.63	0.14	20.87	1	<0.001	1.88	1.44–2.47	1.89	1.44–2.48
Metastasis	0.41	0.13	9.97	1	0.002	1.50	1.17–1.94	1.51	1.17–1.94
Constant	−2.35	0.10	530.54	1	<0.001	0.09		0.12	
ESPEN ^a^	0.07	0.26	0.08	1	0.772	1.08	0.65–1.79	1.08	0.65–1.79
Patient type	0.75	0.13	31.99	1	<0.001	2.12	1.63–2.75	2.12	1.63–2.75
Metastasis	0.55	0.12	20.06	1	<0.001	1.74	1.37–2.22	1.74	1.37–2.22
Constant	−2.28	0.10	530.11	1	<0.001	0.10		0.12	
ICD-10 ^a^	0.45	0.17	7.12	1	0.008	1.57	1.13–2.20	1.59	1.14–2.21
Patient type	0.69	0.14	24.46	1	<0.001	1.99	1.52–2.62	2.00	1.52–2.63
Metastasis	0.53	0.13	17.31	1	0.001	1.70	1.32–2.17	1.70	1.32–2.18
Constant	−2.33	0.10	525.20	1	<0.001	0.10		0.12	

Abbreviations: GLIM, Global Leadership Initiative on Malnutrition criteria; ESPEN, European Society of Clinical Nutrition and Metabolism malnutrition criteria; ICD-10, International classification of disease version 10 malnutrition criteria; CI, confidence interval. All final models contained three independent variables: patient type, the presence of metastasis and the chosen malnutrition diagnostic criteria. ^a^ The response options for variables were GLIM, ESPEN and ICD-10 malnourished versus not malnourished (reference group), patient type; inpatient versus ambulatory (reference group), presence of metastatic disease; yes versus no (reference group). ^b^ Adjusted odds ratio for the models including age and sex as covariates.

## Data Availability

Not applicable.

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
