# Peer review of "Comparison of the GLIM, ESPEN and ICD-10 Criteria to Diagnose Malnutrition and Predict 30-Day Outcomes: An Observational Study in an Oncology Population"

_nutrients, 2021, doi:10.3390/nu13082602_

Round 1

Reviewer 1 Report

Thank you for giving me the opportunity to review this article. The current manuscript entitled “Comparison of the GLIM, ESPEN and ICD-10 criteria to diagnose malnutrition and predict 30-day outcomes: an observational study in an oncology population” is a well-written article. It compares the prevalence of malnutrition using the Global Leadership Initiative on Malnutrition (GLIM), the European Society for Clinical Nutrition and Metabolism (ESPEN) and the International Statistical Classification of Diseases version 10 (ICD-10) criteria in an oncology population. In addition, it compares the level of agreement between these criteria and aimed to identify the predictive validity of each criterion with respect to 30-day outcomes.

-The major concern of the article is in Table 4. I strongly recommend adjusting the model for gender, age and primary malignancy. The results may be overestimated. I am curious to know the R2 of the model, as the authors use the model to predict mortality and unplanned admission or readmission.

-The image of Figure 1 is moved.

Author Response

Please refer to response in the attached word document.

Reviewer 2 Report

This study aims to investigate and compare the prevalence of mal-nutrition using the GLIM, European Society for Clinical Nutrition and Metabolism (ESPEN) and International Statistical Classification of Diseases version 10 (ICD-10) criteria, compare the level of agreement between these criteria, and identify the predictive validity of each criteria with respect to 30-day outcomes in a large cancer cohort of 2794 participants.

The authors concluded that slight to fair agreement was reported between the criteria and that all three criteria were predictive of mortality, but only the GLIM and ICD-10 criteria were predictive of unplanned admission or readmission at 30-days.

This article is well written and of clinical interest.

However, several issues should be improved before the consideration for publication.

Major comments

1 In Line 116, who did screen the patients using MST?, which influences the proportion of patients with not malnutrition. What kinds of health professionals? Some specific persons?

2 I would like to know the numbers and proportions of mortality and readmission in each cell of Table 3.

3 It is unclear how to calculate the odds ratios in Table 4.

  Odds ratio is to be calculated in comparison with reference. However, the references of patients type and metastasis are not described. Please add detail explanations in the table caption.

4 In the last of abstract and conclusion in the text, it may be better to add the phrase of, for instance, “in an oncology population”, just in case.

Minor comments

1 In Table 3, the number of 4.9% in ICD-10 CRITERIA Not malnourished may be incorrect.

Author Response

Please refer to the response in the attached word document.

Round 2

Reviewer 1 Report

Thank you for giving me the opportunity to review this article. The current manuscript entitled “Comparison of the GLIM, ESPEN and ICD-10 criteria to diagnose malnutrition and predict 30-day outcomes: an observational study in an oncology population” is a well-written article. It compares the prevalence of malnutrition using the Global Leadership Initiative on Malnutrition (GLIM), the European Society for Clinical Nutrition and Metabolism (ESPEN) and the International Statistical Classification of Diseases version 10 (ICD-10) criteria in an oncology population. In addition, it compares the level of agreement between these criteria and aimed to identify the predictive validity of each criterion with respect to 30-day outcomes.

The authors have appropriately modified the article according the reviewers suggestions. 

Reviewer 2 Report

The manuscript has been improved according to the comments.

Thank you for interesting paper.